# Clinical and prognostic relevance of sST2 in adults with dengue-associated cardiac impairment and severe dengue

**Andrew Teo** [1,2,3☯]*, **Po Ying Chia** [1,3,4☯], **Gaurav Kumar Ramireddi** [5], **Sebastian Kah Ming Khoo** [1], **Tsin Wen Yeo** [1,3,4]*

**1** Lee Kong Chian School of Medicine, Nanyang Technological University, Singapore, Singapore, **2** Department of Medicine, The Doherty Institute, University of Melbourne, Melbourne, Australia, **3** National Centre for Infectious Diseases, Singapore, Singapore, **4** Department of Infectious Diseases, Tan Tock Seng Hospital, Singapore, Singapore, **5** Department of Medicine, Monash University, Clayton, Australia

☯ These authors contributed equally to this work.
* andrewcc.teo@ntu.edu.sg (AT); yeotsinwen@ntu.edu.sg (YTW)

**Data Availability Statement:** All relevant data are within the manuscript and its Supporting Information files.

## Abstract

### Background

Dengue can be complicated by severe outcomes including cardiac impairment, and the lack of reliable prognostic biomarkers poses a challenge in managing febrile dengue patients. Here, we investigated the functionality of soluble suppressor of tumorigenicity (sST2) as a predictive marker of severe dengue and its association in dengue-associated cardiac impairment.

### Methods

Plasma samples, aged >16 years, collected from 36 dengue fever, 43 dengue with warning signs, 11 severe dengue (collected at febrile, critical and recovery phases) and 30 controls were assayed for plasma levels of sST2, troponin T and N-terminal (NT)-pro hormone brain natriuretic peptide (NT-proBNP) by ELISA. Cardiac parameters: stroke index (SI), cardiac index (CI) and Granov-Goor Index (GGI) were measured with a bioimpedance device during the different phases for dengue subjects and once for the controls.

### Principal findings

In the febrile, critical and early recovery phases, sST2 levels were significantly elevated in dengue participants and sST2 levels increased with increasing disease severity ($P < 0.01$ for all). sST2 concentrations were negatively correlated with SI ($r = -0.48$; $P < 0.001$, $r = -0.55$; $P < 0.001$), CI ($r = -0.26$; $P = 0.02$, $r = -0.6$: $P < 0.001$) and GGI ($r = -0.44$; $P < 0.001$, $r = -0.57$; $P < 0.001$) in the critical and early recovery phases. In contrast, sST2 levels in the febrile and critical phases, were positive correlated to troponin T ($r = 0.44$, $P < 0.001$; $r = 0.22$, $P = 0.03$, respectively) and NT-proBNP ($r = 0.21$, $P = 0.03$; $r = 0.35$, $P < 0.001$). ROC analysis demonstrated sST2 as a good biomarker of severe dengue in the critical phase, AUROC 0.79, $P < 0.001$.

**Funding:** This study was funded by Clinician Scientist Award INV 15nov007 awarded to TWY and National Centre for Infectious Diseases Catalyst Grant FY202008AT awarded to AT; AT is supported by LKC Medicine Dean's Postdoctoral Fellowship and Nanyang Technological University Research Scholarship Block; PYC was supported by NMRC Research Training Fellowship PYC (NMRC/Fellowship/0056/2018). The funders had no role in study design, data collection and analysis, and decision to publish, or preparation of the manuscript.

**Competing interests:** The authors have declared that no competing interests exist.

## Conclusion/Significance

sST2 levels were elevated in patients with dengue especially in cases of severe dengue. Furthermore, increased sST2 levels were associated with cardiac indicators suggesting lower cardiac performance. While further research is needed to demonstrate its clinical utility, sST2 may be a useful prognostic biomarker of severe dengue.

## Author summary

Dengue is a vector-borne viral disease that infects up to 400 million people a year. It is mostly self-resolving, but in some patients, severe outcomes such as dengue-associated cardiac impairment may occur. The lack of reliable biomarkers for severe dengue poses a challenge in managing febrile dengue patients. Suppressor of tumorigenicity (ST2) is a member of the interleukin-1 receptor family, and its soluble form (sST2) has been a valuable biomarker in patients with heart failure. In a longitudinal cohort of dengue subjects, we investigated the potential utility of sST2 as a predictive biomarker for severe dengue and explored its associations with dengue-associated cardiac impairment. Our work demonstrated sST2 to be significantly raised in dengue participants and levels increase with worse dengue outcomes. We also showed that sST2 levels correlated with lower measurements of cardiac performance, and more importantly, sST2 was observed to be a possible prognostic biomarker of severe dengue. Together, this work provides novel evidence on the potential utility of sST2 to predict severe dengue and its associations in dengue-associated cardiac impairment.

## Introduction

Dengue is a vector-borne viral infection endemic in the tropics and sub-tropics regions. The global incidence of dengue has risen dramatically, and Asia carries the highest disease burden [1]. Dengue is mostly asymptomatic and self-limiting, however, in some patients, severe dengue (SD) may manifest [2].

Dengue can be categorised into febrile (days 1–3), critical/defervescence, (days 4–6) and recovery phases (day 7 onwards), and one of the more prominent features of SD (often in the critical phase), is vascular leakage. Vascular leakage may then lead to severe bleeding, organ-failure and potential fatal circulatory compromise causing shock [3,4]. The cause of dengue shock is multifactorial and cardiac impairment in clinical dengue, which may manifest as myocarditis, myocardial impairment and arrhythmias has been proposed to contribute to pathology [5–8]. While several mechanisms including dysregulated host inflammatory mediators and viral proteins have been hypothesized to mediate dengue shock, the precise mechanism remains unclear [9–11].

Several markers including sickle cell disease (haemoglobin SC and SS), change in haematological parameters (decreased platelets, leucopenia), elevated levels of hepatic enzymes (transaminases, creatine kinase) have been associated with dengue haemorrhagic fever, dengue shock and mortality [12–17]. However, reliable predictive biomarkers of cardiac impairment as predictor of SD are lacking and remains a challenge in the clinical management of febrile dengue patients.

Suppression of tumorigenicity-2 (ST2) is a member of the interleukin-1 receptor group and is expressed as a transmembrane ligand (ST2L) on various immunological cells. Upon inflammatory stimulus, ST2L is released in its soluble isoform (sST2) and acts as a decoy receptor for IL-33 to counterbalance the IL-33/ST2L inflammatory signalling pathway [18]. Interestingly, sST2 has been demonstrated to be a strong, independent predictor of severity and mortality in clinical studies of cardiac failure [18,19]. In younger dengue patients (median age range 3–25 years), sST2 levels were found to be elevated and was associated with disease severity in the acute phase of disease [20–23]. In another study of an elderly dengue cohort, an increase in sST2 levels between the febrile and critical phases was associated with dengue mortality; however, its role as a prognostic biomarker for SD was not evaluated [24]. Additionally, the association of sST2 as a biomarker of dengue-associated cardiac impairment has not been explored.

In the current study, our objectives were to assess the utility of sST2 as a prognostic biomarker for SD and to determine the association between sST2 levels and dengue-associated cardiac impairment. We hypothesized that sST2 levels will be increased in proportion to disease severity and will be associated with dengue-associated cardiac dysfunction in these patients.

## Materials and methods

### Ethics statement

This study was approved by the National Healthcare Group Domain Specific Review Board (E/2016/00982). Written informed consent was obtained from all patients prior to enrolment.

## Participants

Samples from dengue patients, >16 years (with no upper limit to age), were recruited and followed up for an observational study (four time points: febrile, critical, early and late recovery phases) at the National Centre for Infectious Diseases, Singapore from September 2017 to December 2019 [8]. Patients who were dengue NS1 positive (SD Bioline Dengue Duo, Korea) and had a fever history of < 7 days were enrolled. Early recovery phase was defined as post critical phase and up to day 14 whereas late recovery was post day 15. Of note dengue is endemic in Singapore and the population experienced periodic waves of dengue [25]. Concurrently, adults who did not experience febrile episodes two weeks prior to recruitment or did not experience dengue in the past six months were enrolled as controls. Classification of dengue disease severities were done at the end of disease or discharge to account of possible SD occurrence and were assigned based on World Health Organization 2009 classification, with the categories being dengue fever (DF), dengue fever with warning signs (DWS), and severe dengue [4]. We defined the critical phase according to the day with the lowest platelet count, along with the highest haematocrit and defervescence pattern. Pregnant and breastfeeding subjects were excluded from this study.

### Measurement of cardiac parameters with non-invasive cardiac system

Cardiac parameters were assessed at febrile, critical and recovery phases with a Non-Invasive Cardiac System (NICaS) (NIMedical, Israel Advanced Technology Industries, Hertzliya, Israel). This is a whole-body electrical bio-impedance monitoring system which is FDA-approved and has been evaluated in several studies as a reliable method to measure cardiac parameters [26,27]. Briefly, parameters recorded included stroke volume (volume of blood pumped out per beat) and cardiac output (volume of blood pumped out per minute = stroke volume × heart rate) that were adjusted to body surface area to yield stroke index (SI) and cardiac index (CI), respectively. The Granov-Goor Index (GGI), an assessment of left ventricular

function, was also calculated, where a value of <10 is considered sensitive and specific for left ventricular ejection fraction of <55%, which has been found to be below the normal range [28].

### Assays on serological levels of sST2 and biomarkers of cardiac impairment

Plasma concentrations of sST2 (1:40 dilution, R&D systems #DY523B-05), and cardiac biomarkers including troponin T (1:4 dilution, Abcam #ab223860) and N-terminal pro brain natriuretic peptide (NT-proBNP) (1:8 dilution, R&D systems #DY3604-05) were assayed by enzyme-linked immunosorbent assays based on manufacturers' protocols. Serial dilutions were done to generate individual standard curves for the determination of sST2 and cardiac proteins concentrations. All available samples were tested in duplicates and read at 450nm with BioTek Synergy H1.

### Statistical methods

Analysis of variance or Kruskal-Wallis test was performed to determine intergroup differences for parametric or non-parametric continuous variables, respectively, and categorical variables were assessed using $\chi^2$ test. Post-hoc pairwise comparisons were used to compare differences between DF, DWS and SD. Spearman's rank correlation was used to determine correlation coefficients between sST2 and cardiac parameters. Longitudinal multilevel mixed effects linear regression models were used to analyse repeated measures per dengue participant with restricted maximum likelihood estimations. Lastly, area under the ROC curve (AUROC) and corresponding 95% confidence interval (CI) was performed to evaluate the potential of sST2 as a biomarker for severe dengue. Statistical analysis was conducted with Stata version 16 (Stata Corp., College Station, Texas) and graphical representations were plotted with Prism (Graph-Pad version 9). A 2-sided value of $P < 0.05$ was considered significant.

## Results

### Participants

One hundred and twenty subjects, comprising 90 dengue patients (36 dengue fever, 43 dengue with warning signs, 11 severe dengue) and 30 controls were recruited. Their plasma samples were assayed for concentrations of sST2, troponin T and NT-proBNP. Additionally, cardiac parameters including SI, CI and GGI were recorded. At enrolment, baseline characteristics were similar across the different disease severities except a higher proportion of SD cases had recorded history of hypertension (63.6%) and had experienced acute myocardial infarction (9.0%). Additionally, SD subjects tended to experience lengthier hospitalisation (7 days) compared the DF (4 days) and DWS (5 days) groups, **Table 1**. Patients were enrolled at median of 4.5 days after the onset of illness. Of the 11 SD participants, three had dengue shock with two requiring inotropes, one had clinically severe bleeding, three had severe dengue hepatitis and eight had other end-organ impairment such as kidney injury and hemophagocytic lymphohistiocytosis. Note, five patients had at least two complications. No participant experienced clinical myocarditis. There were also no mortalities in our study.

### Levels of soluble suppressor of tumorigenicity (sST2) is proportional to dengue severity

Overall, plasma sST2 levels were significantly higher in dengue patients in the febrile and critical phases, compared to controls, P < 0.001 (**Fig 1A and 1B and Table 2**). In the critical phase, sST2 concentrations were significantly higher in the SD group compared to DF and DWS

**Table 1. Baseline Characteristics of controls, dengue fever, dengue with warning signs and severe dengue patients, and clinical outcomes of dengue participants.**

| Dengue PCR positive, n (%) | 40 (44) | | | | |
|---|---|---|---|---|---|
| Serotypes, n (%) | DENV-1 1 (2.5) | | DENV-2 23 (57.5) | DENV-3 15 (37.5) | DENV-4 1 (2.5) |
| | Controls (n = 30) | DF (n = 36) | DWS (n = 43) | SD (n = 1 1) | $p$ value[a] |
| **Baseline characteristics** | | | | | |
| Male, n (%) | 15 (50.0) | 25 (69) | 25 (58) | 6 (55) | 0.43 |
| Age (IQR), [range], years | 44 (32–59) [23–75] | 43 (29–56) [17–73] | 47 (35–60) [21–80] | 58 (36–66) [24–68] | 0.28 |
| Temperature (IQR), ˚C | 36.6 (36.5–37.0) | 37.6 (37.0–38.2) | 37.6 (36.7–38.1) | 37.7 (36.7–38.6) | **<0.001** |
| BMI (IQR), kg/m$^2$ | 24.3 (21.9–26.4) | 25.2 (22.7–27.8) | 23.5 (21.2–27.8) | 25.1 (22.5–30.1) | 0.42 |
| CCI[b] (IQR), [range] | 0 (0–0) [0–2] | 0 (0–0) [0–3] | 0 (0–1) [0–3] | 0 (0–1) [0–5] | 0.28 |
| AMI, n (%) | 0 | 0 | 1 (2) | 1 (9) | 0.17 |
| Heart failure, n (%) | 0 | 0 | 1 (2) | 0 | 0.6 |
| Heart Rate (IQR), beats/min | 70 (59–75) | 87 (81–94) | 79 (66–87) | 99 (71–100) | **<0.001** |
| Systolic BP (IQR), mmHg | 125 (115–137) | 119 (106–127) | 113 (106–121) | 113 (109–126) | **0.004** |
| Diastolic BP (IQR), mmHg | 74 (66–82) | 66 (60–75) | 67 (59–70) | 68 (60–88) | **0.01** |
| Haematocrit (IQR), % | NA | 42.3 (38.2–45.1) | 40.2 (38.0–43.0) | 42.4 (41.2–46.0) | 0.26 |
| Leukocytes (IQR), × 10$^9$/L | NA | 2.6 (1.9–3.8) | 2.8 (2.3–3.4) | 3.8 (3.2–8.7) | **0.02** |
| Neutrophil (IQR), x 10$^9$/L | NA | 1.67 (0.98–2.60) | 2.04 (1.24–2.41) | 2.60 (1.57–6.86) | **0.02** |
| Platelets (IQR), × 10$^9$/L | NA | 118 (46–149) | 85 (27–140) | 54 (17–93) | **0.039** |
| Hypertension, n (%) | 4 (13.3) | 8 (20.5) | 12 (27.9) | 7 (63.6) | **0.01** |
| Previous dengue[c], n (%) | 3 (10) | 1 (2.8) | 1 (2.3) | 2 (18.8) | 0.13 |
| **Days[d] of clinical symptoms** | | | | | |
| Day of illness at febrile phase (IQR) | N.A. | 4 (3–5) | 4 (3–5) | 5 (4–5) | 0.87 |
| Day of illness at critical phase (IQR) | N.A. | 6 (5–7) | 6 (5–7) | 6 (5.5–7) | 0.28 |
| Day post illness at early recovery phase (IQR) | N.A. | 8.5 (7–10.5) | 8 (7–8) | 8 (7–10) | 0.20 |
| Day post illness at late recovery phase (IQR) | N.A. | 20.5 (16–24.5) | 17.5 (15–21.5) | 18 (15–26) | 0.10 |
| **Hospitalization outcomes** | | | | | |
| Length of hospital stay (IQR), days | N.A. | 4 (3–5.5)[e] | 5 (4–6) | 7 (5–8) | **0.010** |
| ICU admission, n (%) | N.A. | 0 | 0 | 1 (8.3) | 0.140 |

Data are presented in median (interquartile range) or no. (%) of subjects, unless otherwise indicated. *P* values of <0.05 were considered statistically significantly (**in bold**).

Abbreviations: DENV, dengue virus; DF, dengue fever; DWS, dengue with warning signs; SD, severe dengue; IQR, interquartile range; BMI, Body mass index; CCI, Charlson Comorbidity Index; AMI, acute myocardial infarction; BP, blood pressure

[a] ANOVA/Kruskal-Wallis or $\chi^2$ test for comparisons across groups

[b] No recognised cases of severe dengue with co-morbidities such as obesity and diabetes

[c] Dengue serology if timing for serology test was appropriate.

[d] Days after first reported symptoms

[e] Includes 24 participants with dengue fever who were admitted during dengue illness.

groups, P < 0.001 and P = 0.005, respectively (**Fig 1B**). Interestingly, despite clinical recovery, sST2 levels appears to remain elevated in dengue patients compared to controls in the early and late recovery phases, P < 0.01, P = 0.06 (**Fig 1C and 1D** and **Table 2**).

## sST2, a potential prognostic biomarker for severe dengue

In the DF and DWS groups, plasma sST2 concentrations were maximal at the febrile phase and decreased thereafter (**Fig 2A and 2B**). In contrast, in the SD group, sST2 levels remained elevated in the febrile and critical phases and only declined in the recovery phases, suggesting

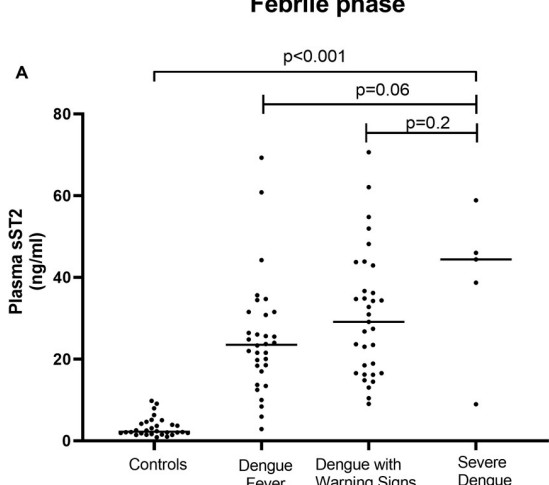

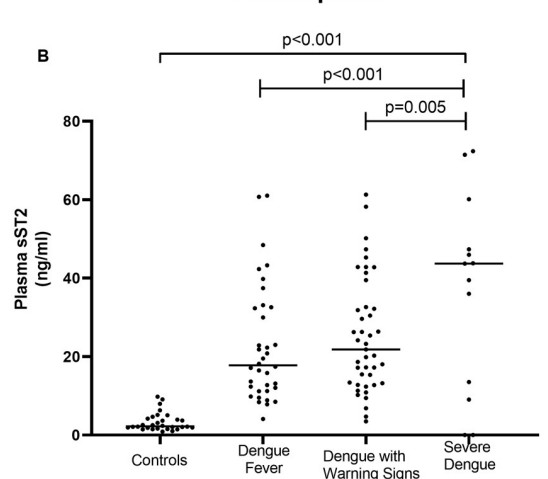

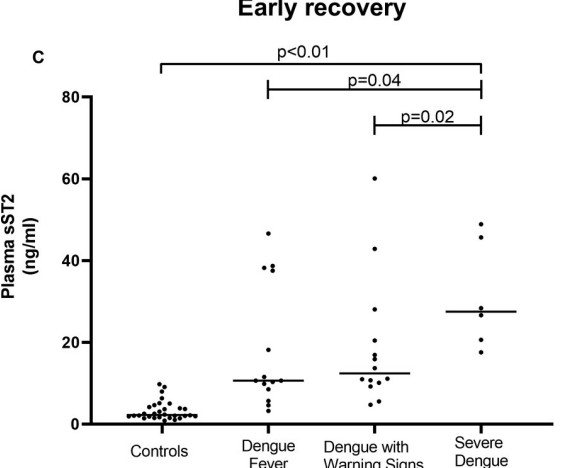

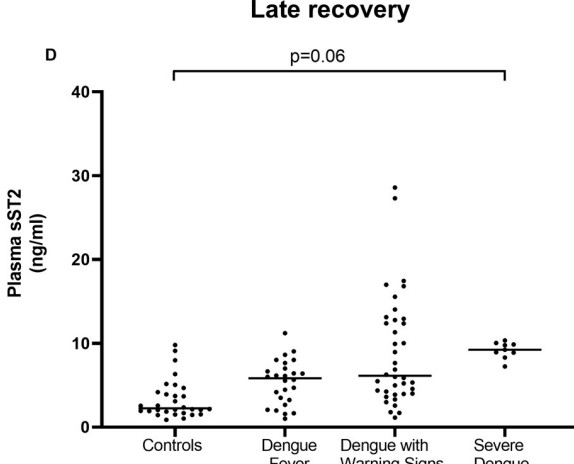

**Fig 1. Plasma levels of sST2 in dengue subjects categorised by disease severities over time.** Dengue subjects (n = 90) classified based on WHO 2009 dengue classifications–dengue fever (n = 36), dengue with warning signs (n = 43) and severe dengue (n = 11); controls (n = 30) collected at one time point were included in the analysis of plasma sST2 levels at: (**A**) early phase; (**B**) critical phase; (**C**) early recovery phase; and (**D**) late recovery phase. Plasma sST2 concentrations were determined by serial dilution of a standard curve. Data are presented as median (horizontal line). P values by Kruskal Wallis tests. Abbreviation: sST2, soluble suppression of tumorigenicity 2.

its usefulness as a biomarker (**Fig 2C**). We then performed AUROC analysis to investigate the effectiveness of sST2 as a prognostic biomarker for SD amongst dengue participants. In the febrile phase, we only had five participants with SD, and the AUROC value for sST2 as a biomarker was 0.74 (95% CI: 0.40–1.00, P = 0.071). Similarly, in the critical phase (SD, n = 11), the AUROC value was 0.79 (95% CI: 0.58–0.97, P = 0.001). We then investigated the effectiveness of sST2 in predicting DWS and SD (n = 38), in the febrile phase, the AUROC was 0.67 (95% CI: 0.54–0.80, P = 0.01). Likewise, DWS and SD patients (n = 62) at the critical phase, the AUROC was 0.64 (95% CI: 0.52–0.75, P = 0.027). We then investigated on the change in sST2 (ΔsST2) levels, (excluding controls), over the course of disease and within the different dengue

**Table 2. Longitudinal comparison of cardiac parameters and plasma cardiac biomarkers in dengue fever, dengue with warning signs, severe dengue patients and controls.**

| | Controls (n = 30) | DF (n = 36) | DWS (n = 43) | SD (n = 11) | p value[a] |
|---|---|---|---|---|---|
| **Stroke Index, mL/m²** | | | | | |
| Febrile phase | 43 (35–47) | 39.5 (36–43) | 38 (30–42) | 32 (31–36) | 0.08 |
| Critical phase | | 35 (32–40) | 32 (27–38) | 30 (27–34) | **<0.001** |
| Early recovery phase | | 35 (32–44) | 36 (28–42) | 28 (25–35) | 0.14 |
| Late recovery phase | | 40 (37–46) | 41 (37–45) | 34 (31–41) | 0.10 |
| **Cardiac Index, L/min/m²** | | | | | |
| Febrile phase | 2.9 (2.6–3.2) | 3.1 (2.5–3.5) | 2.8 (2.2–3.2) | 3.5 (2.8–3.6) | 0.28 |
| Critical phase | | 2.6 (2.3–2.9) | 2.5 (1.9–2.9) | 2.6 (2.2–3.4) | **0.03** |
| Early recovery phase | | 2.7 (2.3–2.9) | 2.5 (2.1–3) | 2.4 (2.1–2.6) | 0.3 |
| Late recovery phase | | 3.0 (2.6–3.5) | 3.0 (2.7–3.4) | 3.0 (2.1–3.3) | 0.36 |
| **Granov Goor Index** | | | | | |
| Febrile phase | 12.9 (11.6–14.6) | 13.2 (10.4–13.8) | 11.6 (9.5–13.4) | 10.5 (9.2–10.5)) | 0.07 |
| Critical Phase | | 11.2 (9.7–12.5) | 10.2 (8.4–11.8) | 9 (8.5–12.2) | **<0.001** |
| Early Recovery Phase | | 11.4 (9.5–13.4) | 11 (8.8–12.3) | 9.2 (8.5–10.8) | **0.041** |
| Late Recovery Phase | | 13.1 (11.2–14.8) | 13.0 (11.6–14.8) | 10.3 (9.3–13.1) | 0.20 |
| **Soluble suppression of tumorigenesis-2, ng/ml** | | | | | |
| Febrile phase | 2.25 (1.85–4.18) | 23.50 (7.69–31.19) | 29.13 (16.55–36.66) | 44.39 (38.70–46.04) | **<0.001** |
| Critical phase | | 17.80 (11.63–32.45) | 21.84 (13.21–32.62) | 43.79 (36.011–60.14) | **<0.001** |
| Early recovery phase | | 10.66 (8.56–37.56) | 11.13 (9.226–20.47) | 27.53 (20.648–45.70) | **0.01** |
| Late recovery phase | | 6.062 (3.51–7.66) | 6.135 (4.00–12.78) | 8.912 (8.60–9.09) | 0.06 |
| **Troponin T, ng/mL** | | | | | |
| Febrile phase | 2.47 (0–4.46) | 6.79 (2.04–9.57) | 5.67 (2.25–7.61) | 2.32 (1.56–8.09) | **0.039** |
| Critical phase | | Below limit of detection (0–1.85) | Below limit of detection (0–0.5) | 2.1 (0–7.1) | **0.03** |
| Early recovery phase | | Below limit of detection (0–0.37) | Below limit of detection (0–0.25) | Below limit of detection (0–0.332) | NA |
| Late recovery phase | | Below limit of detection (0–0.665) | Below limit of detection (0–0.623) | Below limit of detection (0–0.709) | NA |
| **N-terminal pro brain natriuretic peptide, ng/ml** | | | | | |
| Febrile Phase | 1.5 (1.4–1.8) | 2.1 (1.8–3.4) | 1.9 (1.6–3.3) | 1.9 (1.5–3.4) | **0.01** |
| Critical Phase | | 1.9 (1.7–2.1) | 1.8 (1.6–2.0) | 1.9 (1.7–3.4) | **0.001** |
| Early Recovery Phase | | 1.9 (1.6–2.0) | 1.7 (1.5–1.9) | 1.9 (1.7–1.9) | 0.50 |
| Late Recovery Phase | | 2.0 (1.7–2.3) | 2.0 (1.7–2.1) | 1.9 (1.7–2.2) | 0.32 |

Data are median (interquartile range), unless otherwise indicated. *P* values of <0.05 were considered statistically significantly (**in bold**).

[a]ANOVA/Kruskal-Wallis for comparisons across the four groups

groups. At each study visit, a significantly higher degree of ΔsST2 levels [coefficient (95% CI), -7.83 (-9.05–6.62) ng/ml, P < 0.0001] was observed. Within the dengue groups (DF as comparator), we observed significantly higher ΔsST2 levels [5.46 (2.12–8.89) ng/ml, P = 0.001] with increasing disease severity.

We then defined percentage maximal change in haematocrit as the largest difference between the highest haematocrit results during dengue illness and baseline hematocrit, divided by the baseline haematocrit. This indirectly reflects the severity of vascular leakage in the critical phase [4]. In the critical phase, there was a significant association between percentage maximal haematocrit change and sST2 levels (r = 0.5, p < 0.001). In terms of severity of thrombocytopenia, there were also significant inverse associations between platelet counts and sST2 levels in the febrile (r = -0.28, p = 0.01), critical (r = -0.46, p < 0.001) and early recovery (r = -0.63, p < 0.001) phases.

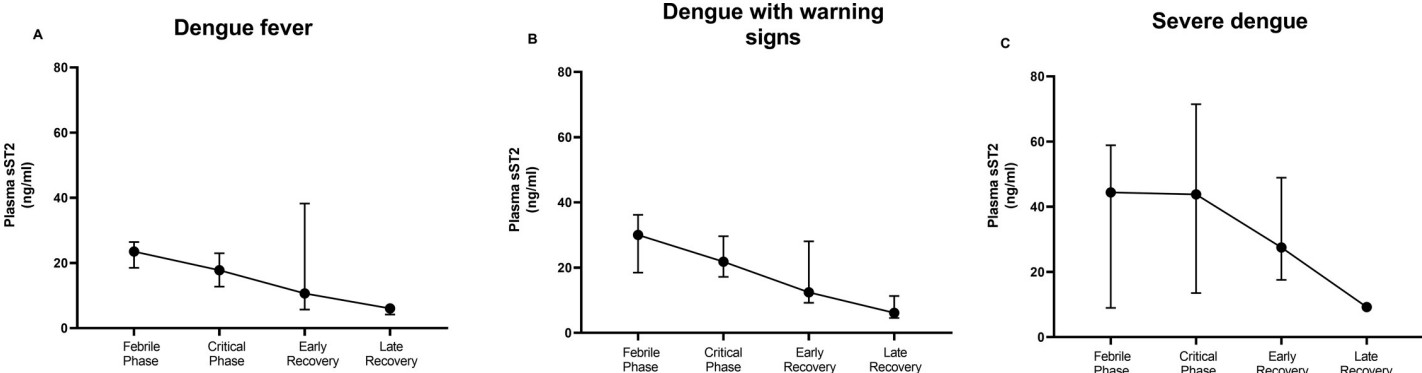

**Fig 2. Plasma levels of sST2 in categorised dengue group at specific time points.** Dengue subjects (n = 90) classified based on WHO 2009 dengue classifications–dengue fever (n = 36), dengue with warning signs (n = 43) and severe dengue (n = 11). (**A**) Dengue fever plasma sST2 concentrations from febrile to late recovery phases. (**B**) DWS plasma sST2 concentrations from febrile to late recovery phases. (**C**) SD plasma sST2 concentrations from febrile to late recovery phases; Plasma sST2 concentrations were determined by serial dilution of a standard curve. Data are presented as median (black circle and 95% CI). Abbreviation: sST2, soluble suppression of tumorigenicity 2; CI, confidence interval; DWS, dengue with warning signs; and SD, severe dengue.

### sST2 levels are correlated with NICaS measured dengue-associated cardiac parameters

Briefly, only in the critical phase, SI, CI and GGI measurements were significantly lower in dengue patients, and we observed a trend of lower readouts as disease severity progress. Longitudinally, these three cardiac parameters decreased from the febrile to critical phase and improved to baseline levels in the late recovery phase. In the current study, there were significant differences in troponinT and NT-proBNP levels in the febrile and critical phases between dengue patients and controls, but not among the SD, DWS and DF groups (**Table 2**).

We then investigated the correlations between sST2 levels and NICaS measurements. In the febrile phase, there were significant inverse correlations between the SI (r = -0.27, P = 0.009) and GGI (r = -0.27, P = 0.008), but not CI. In the critical phase, significant inverse associations between sST2 levels were also seen for SI (r = -0.48, P < 0.001), CI (r = -0.26, P = 0.02) and GGI (r = -0.44, P < 0.001), (**Fig 3A–3C**). Interestingly, these inverse relationships persisted to the early recovery phase: SI (r = -0.55, P < 0.001), CI (r = -0.60, P < 0.001) and GGI (r = -0.57, P < 0.001 (**Fig 3E and 3F**). In contrast, sST2 levels in the febrile and critical phases, were positively correlated to troponin T (r = 0.44, P < 0.001; r = 0.22, P = 0.03, respectively) and NT-proBNP (r = 0.21, P = 0.03; r = 0.35, P < 0.001, respectively). Lastly, there was no association between SI, CI and GGI with troponin T or NT-proBNP in any of the dengue phases.

## Discussion

Dengue is highly prevalent in Asia, and approximately 19.1% of febrile dengue patients require hospitalisation, which can overwhelm health care systems during dengue transmission season [29]. Although dengue is often self-limiting, SD may manifest in some individuals. The lack of reliable predictive biomarkers of SD poses a major challenge in early diagnosis of SD. We showed that in our cohort of dengue subjects, sST2 protein concentrations were elevated in proportion to disease severity during the febrile, critical, and early recovery phases, and sST2 was predictive of SD in the critical phase. Furthermore, in multilevel regression model, the kinetics of sST2 increased with increasing disease severity, Lastly, sST2 levels was inversely correlated with NICaS measured cardiac parameters and positively associated with established serological markers of cardiac impairment in the critical phase.

## Critical phase

## Early recovery

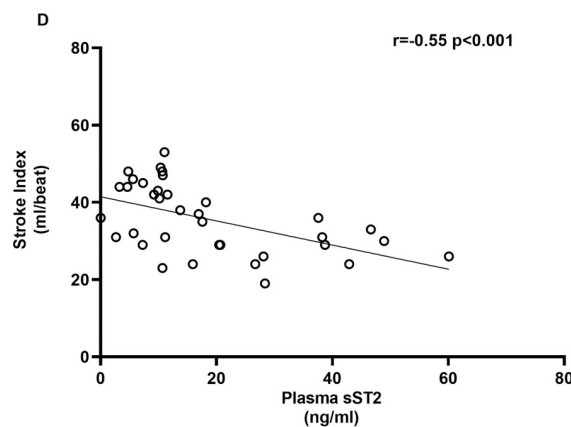

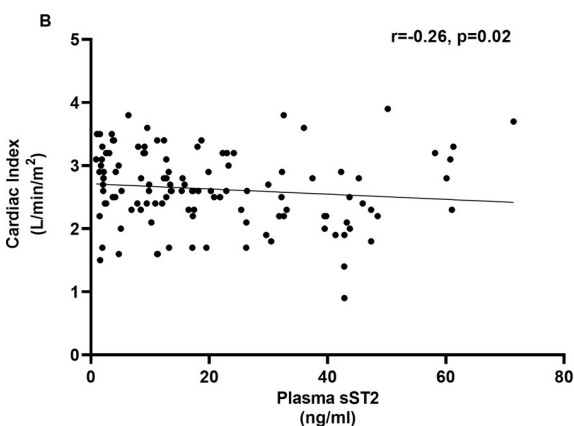

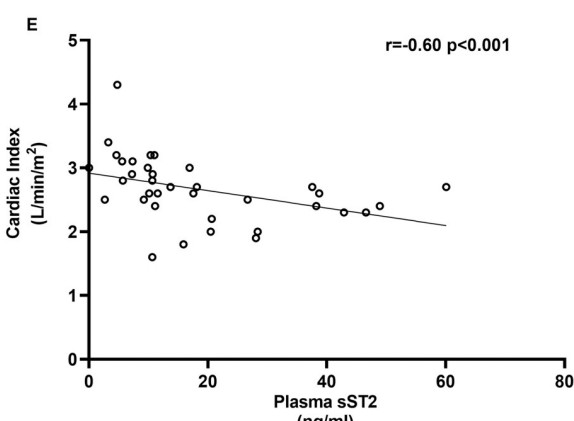

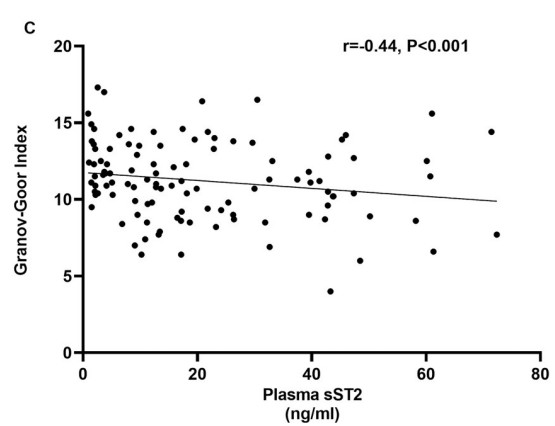

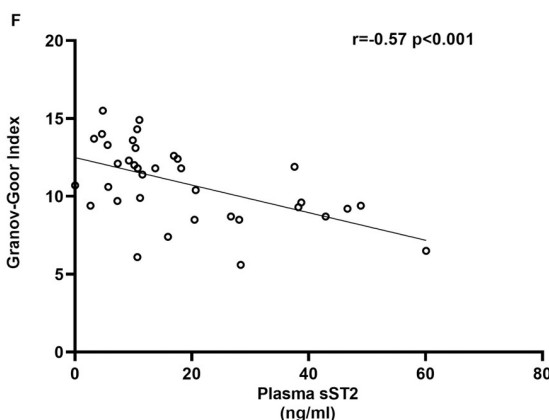

**Fig 3. Correlations between plasma sST2 levels and NICaS measure cardiac parameters at critical phase and early recovery.** Dengue subjects (n = 90). (A-C) Critical phase and (D-F) Early recovery phase, dengue plasma sST2 levels correlations with (A, D) stroke index; (B, E) cardiac index; and (C, F) Granov-Goor index. Plasma sST2 concentrations were determined by serial dilution of a standard curve and cardiac parameters measure with whole body bioimpedance device Data presented as dot plots. P values by spearman correlations with R values. Abbreviation: sST2, soluble suppression of tumorigenicity 2; NICaS, Non-Invasive Cardiac System.

Our observation of increased serological sST2 levels in our cohort of adult dengue patients mirrors earlier studies comprising of younger adults and children [20–23]. The higher sST2 concentrations in proportion to disease severity reported here, may indicate sST2 involvement in SD. In support, sST2 levels were observed to be positively correlated with maximal haematocrit change and negatively correlated with platelet counts, suggesting its association with vascular leakage [20]. Increased sST2 levels may indirectly suggest higher inflammation contributing to severe pathology. One proposed mechanism of SD is the dysregulation of inflammation termed "cytokine storms", and earlier observations of positive associations between sST2 and IL-6-and-8 supports this [22]. Furthermore, in ST2 knocked out mice, dengue pathologies were less severe compared to wild-type mice, and IL33 was proposed to mediate dengue severity [30]. Interestingly, despite clinical recovery, sST2 levels appeared to remain elevated in the early and late recovery phases compared to controls, especially in the SD group. Whether the observations reflect prolonged inflammation or potential long-term sequalae of dengue require further long-term studies.

In the SD group, plasma sST2 levels in the febrile phase (median day 5 of illness) was higher across the different dengue categories and sST2 levels remained elevated in the critical phase, which was not evident in either the DF or DWS groups. This agrees with a recent study from a cohort of elderly Taiwanese demonstrating elevated serum sST2 levels (median day 6) but not earlier time point (median day 2) to be associated with SD [24]. In the same study, the serial change in sST2 levels during hospitalization was predictive of dengue deaths, AUROC 0.85 (95%CI 0.73–0.98) [24]. Although we did not observe mortalities in our study, nonetheless, we showed significantly increased kinetics of sST2 to be associated with severe disease. Importantly, the AUROC value observed in the present study (0.79) to predict SD in the critical phase, is in line with a paediatric study in Cambodia that utilized combined clinical and laboratory features to predict SD (AUROC, 0.78) [31]. Together, supports the hypothesis of sST2 as a potential biomarker of SD.

Our study is the first to report increased sST2 levels to be implicated with transient decreased in cardiac performance in febrile, critical and early recovery phases. In contrast, no associations with NT-proBNP or troponin T with cardiac parameters were observed. This suggest that in dengue subjects with transient impaired cardiac performance, sST2 may be a more sensitive marker compared to troponin T or NT-proBNP. Of note, a significantly higher proportion of SD subjects had recorded history of hypertension at recruitment, and hypertension is a known risk factor poorer cardiovascular functions [32]. Whether sST2 is an effective prognostic marker of more severe form of dengue-associated impairment should be investigated. A recent mouse model studying dengue-associated cardiac dysfunctions may be useful to map out mechanistic link of sST2 in dengue-associated cardiac impairment [33].

Strengths of the current study include systematic enrolment and longitudinal follow up in the various dengue severity groups. This has enabled longitudinal assessments to define the kinetics of sST2 over the course of dengue and its utility as a biomarker of SD as compared to a single time point. There are a few limitations in the current study, the number of subjects enrolled in in the febrile phase who then progressed to develop SD was small, this limited the capacity to fully evaluate the prognostic value of sST2 in SD. Also, the transient cardiac impairment observed did not correlate with clinically relevant cardiac dysfunction, thus, we

were unable to determine the association between sST2 and more severe form of dengue-associated cardiac impairment. Further studies with larger, more diverse classifications of SD and wider age groups are required to validate the role of sST2 as a biomarker for SD early in the acute febrile phase.

In conclusion, sST2 was increased in adults with dengue compared to controls, and levels were significantly elevated in proportion with disease severity in the critical phase that may persist despite clinical recovery. Importantly, sST2 was a predictor of SD in the critical phase, and elevated sST2 levels was also associated with transient decreased dengue-associated cardiac performance in the febrile, critical and early recovery phases. Further studies are required to further evaluate and validate the use of sST2 as a biomarker to predict SD in the febrile phases.

## Acknowledgments

We acknowledge Shiau Hui Dong, Diana Bee Har Tan and Nadiah Binte Abdul Karim for their work and effort in enrolling the patients, and Jaminah D/O Mohamed Ali for sample processing. We also acknowledge Assistant Professor Chai Ping and Professor Mark Richards for advice on sST2. We would like to appreciate all participants who volunteered their time and effort to take part in the study.

## Author Contributions

**Conceptualization:** Andrew Teo, Po Ying Chia, Tsin Wen Yeo.

**Data curation:** Andrew Teo, Gaurav Kumar Ramireddi, Sebastian Kah Ming Khoo.

**Formal analysis:** Po Ying Chia, Tsin Wen Yeo.

**Funding acquisition:** Andrew Teo, Tsin Wen Yeo.

**Investigation:** Andrew Teo.

**Methodology:** Andrew Teo.

**Resources:** Tsin Wen Yeo.

**Supervision:** Andrew Teo.

**Writing – original draft:** Tsin Wen Yeo.

**Writing – review & editing:** Andrew Teo, Po Ying Chia, Gaurav Kumar Ramireddi, Sebastian Kah Ming Khoo, Tsin Wen Yeo.

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
