## [Decision Letter · Decision Letter 0]

24 Jul 2022

Dear Dr Teo,

Thank you very much for submitting your manuscript "Clinical and prognostic relevance of sST2 in adults with dengue-associated cardiac impairment and severe dengue" for consideration at PLOS Neglected Tropical Diseases. As with all papers reviewed by the journal, your manuscript was reviewed by members of the editorial board and by several independent reviewers. In light of the reviews (below this email), we would like to invite the resubmission of a significantly-revised version that takes into account the reviewers' comments. 

We cannot make any decision about publication until we have seen the revised manuscript and your response to the reviewers' comments. Your revised manuscript is also likely to be sent to reviewers for further evaluation.

Sincerely,

Michael Cappello, MD

Academic Editor

David Harley

Section Editor

Reviewer's Responses to Questions

**Key Review Criteria Required for Acceptance?**

**Methods**

-Are the objectives of the study clearly articulated with a clear testable hypothesis stated?

-Is the study design appropriate to address the stated objectives?

-Is the population clearly described and appropriate for the hypothesis being tested?

-Is the sample size sufficient to ensure adequate power to address the hypothesis being tested?

-Were correct statistical analysis used to support conclusions?

-Are there concerns about ethical or regulatory requirements being met?

Reviewer #1: The objectives could be more clearly stated

The study design is appropriate 

The population though appropriate, could be more clearly described

The sample size is underpowered within certain subgroups

Statistical analysis are appropriate, though additional methods could still be explored

Ethical and regulatory requirements were met

Summary: In this study the authors posit that “sST2 - soluble suppressor of tumorigenicity” was increased in adults with dengue in proportion with disease severity in the febrile and critical phases that may persist despite clinical recovery. Importantly, sST2 was a predictor of severe dengue in the critical phase, and elevated sST2 was also associated with dengue-associated cardiac dysfunction in the febrile, critical and early recovery phases. 

Abstract: Line 3-4: Is the objective of the study to evaluate the utility of soluble suppressor of tumorigenicity (sST2) as a marker of severe dengue? Or is it to evaluate the utility of soluble suppressor of tumorigenicity (sST2) as a marker of cardiac impairment in severe dengue?

Line 15-18; 40-41; 225-226; 251-252: Regarding reference to the “lack of” prognostic biomarkers of severe dengue … is there any published evidence of the many predictive biomarkers of severe dengue? Maybe could summarize the historical evidence thus far in the background literature and mention evidence which certainly exists for children and youth (eg., Hb SC genotype, change in hemoglobin concentration, hepatic transaminitis with elevated LDH and CPK, multiple organ system involvement (etc), which were statistically significant predictors for severe dengue with hemorrhage and attributable mortality (viz: Lue, Frontiers in Medicine, 2022; Rankine-Mullings et al, Ebiomedicine, 2015; Elenga et al, J Infect Public Health, 2020; Kularatman et al, BMC Pediatrics, 2019; Sani, BMC Infectious Diseases, 2017; Prasad, PIDJ, 2020) and then say … However, prognostic biomarkers of cardiac malfunction as predictors of severe dengue need to be determined. In this study we sought to …

Background: What was the prevalence of dengue in the community when and where the study was being performed?

58-61 – Dengue shock syndrome (DSS). Have the terminology and criteria been since revised (refs # 3-7 now updated to WHO’s ref #17)?

Reviewer #2: The objectives of the study and the authors' hypotheses are clearly articulated. 

The participants section of the Methods could benefit from additional details. What were the ages you intended to recruit? Did you have a specific timeframe for fever duration to be able to participate (e.g. 7 days or less)? Was the disease severity classification done at admission, or at the end of the disease? 

It seems that 24/36 dengue without warning signs patients in this study were hospitalized, 2/3 seems a little high as hospitalization is not typically warranted for these patients. Were 24 patients hospitalized as dengue without warning signs? Did they have additional risk factors that granted hospitalization? Was the hospitalization later in the disease because the disease severity increased? You mention in the document that none of the patients had cardiac compromise, but it would also be beneficial to include information on the specific type of severe dengue for the 11 patients included in this group (shock vs bleeding vs organ compromise). Did any patient die?

(The information included in Line 278 regarding enrollment should be included in the methods and results)

A figure explaining the recruitment and follow-up process might be helpful to better understand the study procedures, including if DENV PCR testing was used, if you had information on DENV serotypes, and how previous dengue status reported in Table 1 was assessed, would also be useful to better define your population.

Reviewer #3: 1) Abstract, line 26 – “p < .001 for all” does not appear to be in accord with Table 2. 

2) Abstract, lines 35-36 – Suggest changing conclusion to something like “While further research is needed to demonstrate its clinical utility, sST2 may be a useful prognostic biomarker of severe dengue.”

3) Author summary, lines 47-49 – Consider changing “poorer cardiac functions” to “transiently lower measurements of cardiac function” and replacing “good prognostic biomarker” to “possible prognostic biomarker.”

4) Introduction, line 71 – Based on references 12 and 13, the words “proportionate to” should be replaced by “associated with” to be accurate and precise. 

5) Materials and Methods, lines 82-83 – Please describe how the study defined early and late recovery phases. 

6) Materials and Methods, lines 90-91 – Did all three criteria for have to be fulfilled in defining the critical phase? How were these determinations made?

**Results**

-Does the analysis presented match the analysis plan?

-Are the results clearly and completely presented?

-Are the figures (Tables, Images) of sufficient quality for clarity?

Reviewer #1: The analysis presented matches the plan

The results are clear

The numbers in the tables could be clarified

The figures are impressive

Results: What was the dengue serotype of patients? What serotypes were circulating in the community? Was ZIKA circulating, also?

Lines 11, 128, Table 1: How many were severe dengue with hemorrhage? Would they have worse cardiac prognostic indicators than those without hemorrhage? Was there a statistical difference? Or were numbers when further disaggregated too small for meaningful comparisons?

Line 132 – correction myocardial “infarction”

Table 1: Some of the numbers (mean? median?) and figures in the brackets (###) are not clear. 

Tables 1, 2; Lines 280-282: With only one with clinical heart failure? one admitted to ICU? No cases of myocarditis? Pericarditis? Echocardiography anomalies? Or deaths? What then is the “clinical correlation” with laboratory results obtained for significant cardiac compromise reported here?

Figures 1,2,3: Trends look impressive

Given multiple potential confounders, since there is a control population, should logistic regression modeling be done to factor in whether other variables may have been affecting outcomes?

Reviewer #2: The results match the analysis plan. 

Line 131: Do you mean they had history of hypertension, or they actually had high blood pressure during clinical assessment? 

Line 134: Consider to include the range of days post-onset of illness at enrolment

Tables: 

Please revise your tables' titles to make sure that they are complete. The tables should be stand-alone.

Table 1: How was previous dengue assessed? 

Table 1: For the days of clinical symptoms, do the presented days correspond to the duration of each phase in number of days? How did you define early and late recovery phase? 

Table 2: Providing normal/expected values for indexes would be helpful

Reviewer #3: Table 2 present interesting data that describe several measures of cardiac function and cardiac biomarkers kinetics during the four illness phases and stratified by disease severity. These are complex data that show a dynamic cardiovascular physiology during the natural history of dengue. This manuscript would benefit from the authors’ viewpoint on why the cardiac index in dengue patients starts higher than controls during the febrile phase (though not statistically significant) and then proceeds to decrease during critical and early recovery phase before returning to values close to controls during late recovery. The same line of thinking could also be applied to changes in the troponin T and NT-proBNP levels. While several analyses of differences among the 4 groups are presented, an analysis of their ability to distinguish severe disease from DF or DWS (especially during the febrile phase) would help the reader evaluate their prognostic value.

The main message of Figure 3 is that sST2 correlated with several of the measures of cardiac function (although I would suggest showing all data, especially during the febrile phase, even if not significant. However, if these measures of cardiac function are not predictive of disease severity, then it does not reinforce the argument that sST2, as a surrogate marker for these measurements, would assist in predicting severity. If data are available of the clinical relevance of these cardiac parameters to the clinical picture, these should be included to justify that a surrogate marker would be of clinical utility.

5) Results, line 131-132 and Table 1 

Data presented in Table 1 do not clearly indicate that 67% of severe dengue cases were baseline hypertensive and the myocardial infarction percentage in the table does not match the text. 

How was previous dengue infection determined as shown in Table 1?

Days of clinical symptoms – The Table description is unclear. Are the days shown the full range (not an interquartile range) for each illness phase. Presumably blood samples were taken during the days indicated. 

Hospital outcomes – Please describe the 1 ICU admission relevant cardiac conditions, if any. Did the patient require inotropies/vasopressors and/or cardiac pacing?

6) Results, Table 2 – In the p value column, please indicate whether the comparison is all 4 groups or comparing controls to all dengue cases combined.

7) Figure 2 – Graphs A and B have the same Y axis scale (0-40) and C (0-80). Please standardize axes for easier comparison. 

8) Results, line 171 – The sentence as written is unclear. The authors seem to be saying that for the 5 severe disease patients with febrile phase samples, the AUROC value was 0.74.

6) Results, lines 172-176 – What is being compared to give p value results for AUROCs is unclear. 

7) Results, lines 177-183 - What was the timing of the sST2 elevation in relation to that of increased hematocrit and drop in platelet count?

8) Results, line 196 - Are you describing differences between controls to all dengue cases combined (comparison between 2 groups) or differences among all 4 groups? Table 2 appears to be comparing among all 4 group which should be precisely worded in these results.

9) Results, line 209: Higher troponin had an inverse relationship to disease severity according to table 2 in the febrile phase. Also, nt-probnp was significantly different between the dengue (unclear if combined dengue or stratified by severity analysis) and controls in the febrile and critical phase, but did not appear to have any difference among dengue severity (an analysis that should be shown, even if not significant).

10) Results - The authors should make clear whether any of the severely ill patients developed shock and that none of their study patients died. Did any of the non-dengue controls have a severe illness?

**Conclusions**

-Are the conclusions supported by the data presented?

-Are the limitations of analysis clearly described?

-Do the authors discuss how these data can be helpful to advance our understanding of the topic under study?

-Is public health relevance addressed?

Reviewer #1: In this study the authors posit that “sST2 - soluble suppressor of tumorigenicity” was increased in adults with dengue in proportion with disease severity in the febrile and critical phases that may persist despite clinical recovery. Importantly, sST2 was a predictor of severe dengue in the critical phase, and elevated sST2 was also associated with dengue-associated cardiac dysfunction in the febrile, critical and early recovery phases. 

In discussing immunological mechanisms for dengue pathogenesis, is there a role for mentioning antibody dependent immune enhancement (ADE), which has been linked to severe dengue, after subsequent infection with another heterogenous dengue serotype, dengue type 2 serotype disease occurring after ZIKA infection; Dengvaxia vaccine (Soo K-M, PLOS One, 2016; Pawitan, Acta Med Indones, 2011; Stettler et al, Science, 2016; Katzelnick et al, Science, 2020; George et al, Sci Rep 2017; Sanofi, 2019)? Are/were India and also Indonesia – where this study was performed, undergoing a significant dengue and/or ZIKA surge? Although controversial, should ADE be mentioned and referenced, also? 

Limitations: Would say in tropical and subtropical developing countries where severe dengue occurs, is it likely that these investigations would be useful to evaluate patients during an epidemic (eg., in Indonesia), when health systems are likely to be overwhelmed and severe cases of hemorrhagic dengue with high attributable morbidity are being encountered? Would you say this is a “pilot study”, given very small numbers overall, small proportions of cases to controls, imbalanced co-morbidities between cases (SD) and controls, lack of clinical correlations, given single cases of admission to ICU and cardiac failure with no cases of myocarditis, pericarditis, or deaths reported? 

Public health relevance could be better outlined/stressed.

Reviewer #2: The discussion section can benefit from additional edits, avoiding too much repetition of the information already presented in the introduction and the results. 

Line 268: It is not completely clear in the methods and results, what do you describe as "cardiac dysfunction". To get to this conclusion, a better description of what is the definition of cardiac dysfunction is needed. Additionally, it would be important to include if any adjustment by other variables that can explain cardiac dysfunction in these patients was made. 

Limitations of the study are clearly stated.

Please consider to include additional discussion on the logistical/technical/economic aspects of using sST2 in clinical settings, as most areas affected by dengue are developing countries (is this an easy test to do? requires expensive supplies/devices? it is expensive? results can be obtained quickly? etc)

Reviewer #3: I would caution the authors’ from using the term cardiac dysfunction (line 268 and 288) to describe the correlation to the noninvasive measurement of cardiac function since they, for the most part, stay above 2.5 which would be considered the normal range for the cardiac index. The clinical significance of these measurements could be strengthened if data were available showing that these low normal cardiac indices resulted in signs/symptoms of heart failure, such as lower extremity edema or pulmonary edema, or sequelae of low cardiac output state leading to tissue hypoperfusion like elevated lactic acid or cyanotic/cool extremities. Other cardiac outcomes such as need for inotropic/vasopressor support or major cardiac assist devices or death due to cardiovascular complications should be presented if these are available. Without these data, these values could be argued to represent normal cardiovascular physiology that is compensating for a dengue disease state. 

11) Discussion, lines 229-231 – Consider refining this statement. Your data show that sST2 was inversely correlated with some cardiac measurements at some time points in disease course, but not consistently. And also, not during the febrile phase when it might be most helpful in guiding treatment. Also, while it was correlated with some cardiac biomarkers, the cardiac biomarkers themselves were not necessarily correlated with cardiac measurements and, in turn, those cardiac measurements were not necessarily correlated with dengue severity.

12) Discussion, lines 234-236 – The authors should note that there was a trend towards higher sST2 levels with disease severity, but not necessarily statistical significance at all time points. 

13) Discussions, lines 241-244 - Unless these data are placed in supplementary material, consider leaving this out, since the readers cannot easily evaluate these data for themselves. Additionally, if you will be including this statement, please place in the results.

14) Discussion, lines 247-248 - In Figure 2, it appears that sST2 levels go down as the disease progresses for all 3 disease severity states. When you state that to “sST2 levels appeared to remain elevated in the early and late recovery phases, especially in the SD group” are you referring to a specific threshold by which they remain elevated?

15) Discussion, lines 252-255 - If one is trying to argue that sST2 is important for predicting progression to SD, the best data are those that focus on comparing 3 dengue severity categories. However, sST2 was not significantly higher in severe dengue compared to dengue fever and dengue with warning signs in the febrile phase. This was only true in the critical and early recovery phase.

16) Discussion, line 258 - 0.79. But 0.78 in the results.

17) Discussion, lines 267-270 - sST2 levels were only associated with cardiac measurements in the critical and early recovery phase. You reported this as not associated with some measurements in the febrile phase. Also, none of the data presented show cardiac measurements (either in SI or CI) that were in the pathologic range or that this could be considered cardiac dysfunction by a clinical measure. Consider rephrasing for as "transiently lower measurements of cardiac function in patients with dengue" as they were not clinically significant.

18) Discussion, line 277 - “utility as a biomarker of SD as compared to a single time point.” Perhaps one should say “possible biomarker of SD when taken at specific time points” since the authors did not analyze change in sST2 levels taken as serial measurements in the same patient as a prognostic factor for dengue. 

19) Discussion, lines 285–286 -- Consider rewriting the conclusion given that sST2 was only statistically significantly increased by disease severity in the critical and early recovery phases.

120) Line 288: See general comments for using “cardiac dysfunction.”

**Editorial and Data Presentation Modifications?**

Reviewer #1: Minor revision

Reviewer #2: Line 20: I think the total samples was 120, as listed below in line 127, but here you list 122. Please check.

Line 16 and Line 40: Use "biomarkerS" instead of "biomarker"

Line 22: NT-probBNP should be spelled out here

Line 49: Not sure the last part of this sentence can be claimed with the authors' findings

Line 52: Dengue is mostly asymptomatic, suggest to revise the language

Line 55-64: Introduction section would benefit for a more detailed explanation of how severe dengue manifests to include severe bleeding and organ involvement, besides shock. 

Line 63: Please consider to use the word "critically'' more sparingly throughout your document

Line 63: Consider to include that many biomarkers have been studied and suggested as potential predictive biomarkers of severity in dengue 

Line 69: Do you mean "in" instead of "and"?

Line 86: Please check language, at it is written now it looks like you are saying that patients who had dengue in the previous 6 months were enrolled as controls. Is that what you mean?

Line 93: You mention "each study visit" but it is not clear from the methods description how this visits occur? Does this mean not all patients were hospitalized? Or "visit" refer to each encounter with the researchers?

Line 101: Please consider to add an explanation of hoy you are using a cut off of <10 or EF<55%, for those not familiar with what this means

Line 170: I cant find figure 2D in the document, is this 2C?

Line 227: Lower age range seems to be 29 by table 1, not sure your population corresponds to older adults

Line 253: Do you mean significantly higher when compared to the other dengue clinical classifications?

Reviewer #3: Lines 20–21: Consider “plasma samples collected from 36 patients with dengue fever, 43 patients…”

Line 34: “sST2 levels were raised in...” Consider changing to “were elevated in patients with dengue.”

Line 42: “valuable biomarker in patients…” consider specifying outcomes for which it was predictive (e.g. mortality, readmission, etc.).

Line 53–54: Although severe dengue can occur with any case of dengue, references to WHO guidelines on early recognition of warning signs and low mortality with appropriate treatment should be referenced.

Lines 59–64: These sentences should be revised to focus on either dengue shock syndrome, severe dengue, or explain the relationship of dengue shock syndrome to severe dengue when switching terms.

Lines 171–172: Is this discriminating SD from DF and DWS combined? Please explicitly describe what sST2 is discriminating between.

Line 175: “DWS and SD patients” – Please put an n if you put one for all other AUROC analyses.

Lines 199–201: “In the current study, there were significant differences in troponin-T and NT-proBNP levels in the febrile and critical phases between dengue patients and controls, but not among the SD, DWS and DF groups” - This test statistic is not shown in the table. If reported as not significant here, it's helpful to still report the p-value in the text.

Lines 260–263: Interesting finding from that paper, but the data presented in this paper did not look at change in sST2 levels, so it’s difficult to make that comparison. Consider removing.

Lines 280–282: Consider adding the limitation that there are no clinical cardiac data that are correlated with the measurements of cardiac output. This is a larger limitation than simply not having myocarditis cases.

Editorial comments

1) Abstract, line 24 – Suggest “… during the different phases for dengue subjects…”

2) Abstract, line 26 – sST2 typo error as sS2

3) Author summary, line 40, and Discussion line 225 – Suggest delete “critically,” a word consistently overused in the manuscript, “The lack of a reliable biomarker…”

4) Author summary, line 43 – Suggest “…we investigated the potential utility of sST2…”

5) Author summary, line 48 – Suggest “.. this work provides novel evidence on the potential utility… 

6) Introduction, line 57 – Suggest shorten to “…features of SD, …, is vascular leakage that may lead to potential fatal cardiovascular collapse…”

7) Line 58: Consider “cardiovascular collapse” should be “circulatory compromise” or “circulatory collapse”

8) Introduction, line 63 – Suggest delete “Critically,” 

9) Line 128: “recruited, and their plasma were…” Consider breaking into two samples and changing language to “Their plasma samples were assayed…”

10) Results, line 131 – “67%” belons after “hypertensive”

11) Results, line 132 – “infarction” missplelled

12 Results, line 146 – Suggest delete “Critically,” 

13) Results, line 170 – Appears that Fig 2D should be Fig 2C. 

14) Results, lines 196-7 – Suggest “…GGi measurements were significantly lower in severe dengue patients only in the critical phase.” 

15) Results, line 203 – Typo, should be “…there were...”

16) Discussion, line 226 – Suggest delete “In here,”

17) Discussion, line 245 – Appears that should read “…less severe in wild-type mice,…”

18) Discussion, line 251 – Should read “Biomarkers to predict SD are lacking…”

19) Discussion, line 257 – Suggest replace “Critically” with “Importantly”

**Summary and General Comments**

Reviewer #1: Summary: In this study the authors posit that “sST2 - soluble suppressor of tumorigenicity” was increased in adults with dengue in proportion with disease severity in the febrile and critical phases that may persist despite clinical recovery. Importantly, sST2 was a predictor of severe dengue in the critical phase, and elevated sST2 was also associated with dengue-associated cardiac dysfunction in the febrile, critical and early recovery phases. 

Abstract: Line 3-4: Is the objective of the study to evaluate the utility of soluble suppressor of tumorigenicity (sST2) as a marker of severe dengue? Or is it to evaluate the utility of soluble suppressor of tumorigenicity (sST2) as a marker of cardiac impairment in severe dengue?

Line 15-18; 40-41; 225-226; 251-252: Regarding reference to the “lack of” prognostic biomarkers of severe dengue … is there any published evidence of the many predictive biomarkers of severe dengue? Maybe could summarize the historical evidence thus far in the background literature and mention evidence which certainly exists for children and youth (eg., Hb SC genotype, change in hemoglobin concentration, hepatic transaminitis with elevated LDH and CPK, multiple organ system involvement (etc), which were statistically significant predictors for severe dengue with hemorrhage and attributable mortality (viz: Lue, Frontiers in Medicine, 2022; Rankine-Mullings et al, Ebiomedicine, 2015; Elenga et al, J Infect Public Health, 2020; Kularatman et al, BMC Pediatrics, 2019; Sani, BMC Infectious Diseases, 2017; Prasad, PIDJ, 2020) and then say … However, prognostic biomarkers of cardiac malfunction as predictors of severe dengue need to be determined. In this study we sought to …

Background: What was the prevalence of dengue in the community when and where the study was being performed?

58-61 – Dengue shock syndrome (DSS). Have the terminology and criteria been since revised (refs # 3-7 now updated to WHO’s ref #17)?

Results: What was the dengue serotype of patients? What serotypes were circulating in the community? Was ZIKA circulating, also?

Lines 11, 128, Table 1: How many were severe dengue with hemorrhage? Would they have worse cardiac prognostic indicators than those without hemorrhage? Was there a statistical difference? Or were numbers when further disaggregated too small for meaningful comparisons?

Table 1: Some of the numbers (mean? median?) and figures in the brackets (###) are not clear. 

Tables 1, 2; Lines 280-282: With only one with clinical heart failure? one admitted to ICU? No cases of myocarditis? Pericarditis? Echocardiography anomalies? Or deaths? What then is the “clinical correlation” with laboratory results obtained for significant cardiac compromise reported here?

Figures 1,2,3: Trends look impressive

Given multiple potential confounders, since there is a control population, should logistic regression modeling be done to factor in whether other variables may have been affecting outcomes?

In discussing immunological mechanisms for dengue pathogenesis, is there a role for mentioning antibody dependent immune enhancement (ADE), which has been linked to severe dengue, after subsequent infection with another heterogenous dengue serotype, dengue type 2 serotype disease occurring after ZIKA infection; Dengvaxia vaccine (Soo K-M, PLOS One, 2016; Pawitan, Acta Med Indones, 2011; Stettler et al, Science, 2016; Katzelnick et al, Science, 2020; George et al, Sci Rep 2017; Sanofi, 2019)? Are/were India and also Indonesia – where this study was performed, undergoing a significant dengue and/or ZIKA surge? Although controversial, should ADE be mentioned and referenced, also? 

Limitations: Would say in tropical and subtropical developing countries where severe dengue occurs, is it likely that these investigations would be useful to evaluate patients during an epidemic (eg., in Indonesia), when health systems are likely to be overwhelmed and severe cases of hemorrhagic dengue with high attributable morbidity are being encountered? Would you say this is a “pilot study”, given very small numbers overall, small proportions of cases to controls, imbalanced co-morbidities between cases (SD) and controls, lack of clinical correlations, given single cases of admission to ICU and cardiac failure with no cases of myocarditis, pericarditis, or deaths reported? 

There are a few minor grammatical errors throughout for correction

Line 132 – correction myocardial “infarction”

Reviewer #2: This manuscript by Teo and colleagues provides additional information on the potential utility of sST2 as a maker of dengue disease severity. This is a very relevant area of research in the dengue field, as there is an urgent need for reliable biomarkers to predict progression to severe disease.

Reviewer #3: The authors present a study of sST2 (soluble suppressor of tumorigenicity, an interleukin-1 receptor group member) as a biomarker for severe dengue in 90 NS1 antigen confirmed dengue adult patients with disease severity classified according to 2009 WHO criteria and 30 non-dengue adult controls in Singapore. Dengue patients had serial quantitative testing of sST2 in each of the febrile, critical, early recovery and late recovery disease clinical phases. The study also measured cardiac performance indices and biomarkers of cardiac impairment, Troponin T and NT-proBNP, and correlated these with sST2 levels as well as with maximum hematocrit increases indicative of vascular leakage, and platelet counts. The study observed statistically significantly higher sST2 levels in dengue patients compared to controls, and significantly higher sST2 levels in severe dengue patients compared to dengue fever and dengue warning sign patients during the critical stages (5-7 days post onset). Area under ROC curve analysis suggested sST2 levels could be a useful biomarker for severe disease, particularly in the critical stage.

This study is reasonably rigorous in its assessment of sST2 as a biomarker for severe dengue in adults, particularly in its use of serial measurements in different dengue clinical phases in the study subjects, and acknowledges the study limitation of only 11 severely ill patients including only 5 severe cases identified during the early febrile stage. The study adds to literature regarding sST2 in severe pediatric dengue illness and limited data in adults. A biomarker to identify risk for severe dengue illness early in the course of the disease would be useful as only a small percentage medically attended dengue patients develop severe illness and clinical decompensation can occur rapidly during the critical stage. While the study is useful, the authors also acknowledge in the discussion that further evaluation of sST2 would be needed before adopting routine testing for sST2 in dengue clinical management protocols. For example, the area under ROC curve values for sST2 in this study are not greater than that seen using simple clinical warning signs without lab testing to predict severe disease as studied in Cambodian children in reference 24. 

The study is reasonably well written, although some editing as suggested by “editorial comments” below would be desirable.

PLOS authors have the option to publish the peer review history of their article (what does this mean?). If published, this will include your full peer review and any attached files.

Reviewer #1: No

Reviewer #2: No

Reviewer #3: Yes: Stephen H Waterman
---

## [Decision Letter · Decision Letter 1]

22 Sep 2022

Dear Dr Teo,

Thank you very much for submitting your manuscript "Clinical and prognostic relevance of sST2 in adults with dengue-associated cardiac impairment and severe dengue" for consideration at PLOS Neglected Tropical Diseases. As with all papers reviewed by the journal, your manuscript was reviewed by members of the editorial board and by several independent reviewers. The reviewers appreciated the attention to an important topic. Based on the reviews, we are likely to accept this manuscript for publication, providing that you modify the manuscript according to the review recommendations. 

Sincerely,

Michael Cappello, MD

Academic Editor

David Harley

Section Editor

Reviewer's Responses to Questions

**Key Review Criteria Required for Acceptance?**

**Methods**

-Are the objectives of the study clearly articulated with a clear testable hypothesis stated?

-Is the study design appropriate to address the stated objectives?

-Is the population clearly described and appropriate for the hypothesis being tested?

-Is the sample size sufficient to ensure adequate power to address the hypothesis being tested?

-Were correct statistical analysis used to support conclusions?

-Are there concerns about ethical or regulatory requirements being met?

Reviewer #1: PLOS Neglected Tropical Diseases submission (PNTD-D-22-00741)

Summary: This pilot study has shown that sST2 levels were elevated in patients older than 18 years with dengue especially in cases of severe dengue and increased sST2 levels were associated with cardiac indicators suggesting lower cardiac performance. They conclude that while sST2 maybe a useful prognostic biomarker of severe dengue, further research is needed to demonstrate its clinical utility.

Assessment: The authors have made the relevant requested corrections. The revised manuscript is now suitable for publication, with minor and other edits, viz:

Lines 19-25 and lines 26-33: Suggest separating the methods (line 19-25) from the results/principal findings (lines 26 to 33), under separate subheadings and paragraphs, for clarity.

Line 20: Suggest adding patient age for clarity, viz: “… patients aged >16 years …” 

Line 75: Suggest including “dengue” in this sentence to make the definitive link between “severe dengue, mortality and sST2”. Reference (17-18)

Lines 76-77; line 255: How old were these “younger” patients? What was/were their age range/s? References (19-22)

Line 147: Would spell out the “8 with other end-organ impairment” and how many organ systems (eg., range, median) were involved per patient, for clarity. These could have adversely contributed to the outcomes.

Line 246: Suggest removing “but”

Table 1, Results, Limitations: Given DENV endemicity in this region and the heterogenous serotypes that were co-circulating (44% positive for DENV: 57.5% DENV 2, 35.5% DENV 3 and 2.5% each for DENV 3 and DENV 4; with shift from DENV 2 to DENV 3) and dengue serotype heterogeneity with repeated infections (asymptomatic infections being the vast majority – not shown here) is a definitive risk factor for severe dengue, shouldn’t these results be “spelled out” in the narrative of the results and mentioned in the limitations (as a confounder), as this is a recognized factor risk factor for severe dengue? 

Table 1, Results: Suggest adding a footnote that there were no recognized cases with co-morbidities of obesity, respiratory disease (including asthma), diabetes, renal failure, infectious diseases, as these are recognized risk factors for severe dengue and maybe confounders (like hypertension, already “spelled-out”) for some of the outcomes you have described.

Thanks for the opportunity to review a second reiteration of your manuscript.

Celia DC Christie

2 Sept, 2022

Reviewer #2: (No Response)

**Results**

-Does the analysis presented match the analysis plan?

-Are the results clearly and completely presented?

-Are the figures (Tables, Images) of sufficient quality for clarity?

Reviewer #1: PLOS Neglected Tropical Diseases submission (PNTD-D-22-00741)

Summary: This pilot study has shown that sST2 levels were elevated in patients older than 18 years with dengue especially in cases of severe dengue and increased sST2 levels were associated with cardiac indicators suggesting lower cardiac performance. They conclude that while sST2 maybe a useful prognostic biomarker of severe dengue, further research is needed to demonstrate its clinical utility.

Assessment: The authors have made the relevant requested corrections. The revised manuscript is now suitable for publication, with minor and other edits, viz:

Lines 19-25 and lines 26-33: Suggest separating the methods (line 19-25) from the results/principal findings (lines 26 to 33), under separate subheadings and paragraphs, for clarity.

Line 20: Suggest adding patient age for clarity, viz: “… patients aged >16 years …” 

Line 75: Suggest including “dengue” in this sentence to make the definitive link between “severe dengue, mortality and sST2”. Reference (17-18)

Lines 76-77; line 255: How old were these “younger” patients? What was/were their age range/s? References (19-22)

Line 147: Would spell out the “8 with other end-organ impairment” and how many organ systems (eg., range, median) were involved per patient, for clarity. These could have adversely contributed to the outcomes.

Line 246: Suggest removing “but”

Table 1, Results, Limitations: Given DENV endemicity in this region and the heterogenous serotypes that were co-circulating (44% positive for DENV: 57.5% DENV 2, 35.5% DENV 3 and 2.5% each for DENV 3 and DENV 4; with shift from DENV 2 to DENV 3) and dengue serotype heterogeneity with repeated infections (asymptomatic infections being the vast majority – not shown here) is a definitive risk factor for severe dengue, shouldn’t these results be “spelled out” in the narrative of the results and mentioned in the limitations (as a confounder), as this is a recognized factor risk factor for severe dengue? 

Table 1, Results: Suggest adding a footnote that there were no recognized cases with co-morbidities of obesity, respiratory disease (including asthma), diabetes, renal failure, infectious diseases, as these are recognized risk factors for severe dengue and maybe confounders (like hypertension, already “spelled-out”) for some of the outcomes you have described.

Thanks for the opportunity to review a second reiteration of your manuscript.

Celia DC Christie

2 Sept, 2022

Reviewer #2: (No Response)

**Conclusions**

-Are the conclusions supported by the data presented?

-Are the limitations of analysis clearly described?

-Do the authors discuss how these data can be helpful to advance our understanding of the topic under study?

-Is public health relevance addressed?

Reviewer #1: PLOS Neglected Tropical Diseases submission (PNTD-D-22-00741)

Summary: This pilot study has shown that sST2 levels were elevated in patients older than 18 years with dengue especially in cases of severe dengue and increased sST2 levels were associated with cardiac indicators suggesting lower cardiac performance. They conclude that while sST2 maybe a useful prognostic biomarker of severe dengue, further research is needed to demonstrate its clinical utility.

Assessment: The authors have made the relevant requested corrections. The revised manuscript is now suitable for publication, with minor and other edits, viz:

Lines 19-25 and lines 26-33: Suggest separating the methods (line 19-25) from the results/principal findings (lines 26 to 33), under separate subheadings and paragraphs, for clarity.

Line 20: Suggest adding patient age for clarity, viz: “… patients aged >16 years …” 

Line 75: Suggest including “dengue” in this sentence to make the definitive link between “severe dengue, mortality and sST2”. Reference (17-18)

Lines 76-77; line 255: How old were these “younger” patients? What was/were their age range/s? References (19-22)

Line 147: Would spell out the “8 with other end-organ impairment” and how many organ systems (eg., range, median) were involved per patient, for clarity. These could have adversely contributed to the outcomes.

Line 246: Suggest removing “but”

Table 1, Results, Limitations: Given DENV endemicity in this region and the heterogenous serotypes that were co-circulating (44% positive for DENV: 57.5% DENV 2, 35.5% DENV 3 and 2.5% each for DENV 3 and DENV 4; with shift from DENV 2 to DENV 3) and dengue serotype heterogeneity with repeated infections (asymptomatic infections being the vast majority – not shown here) is a definitive risk factor for severe dengue, shouldn’t these results be “spelled out” in the narrative of the results and mentioned in the limitations (as a confounder), as this is a recognized factor risk factor for severe dengue? 

Table 1, Results: Suggest adding a footnote that there were no recognized cases with co-morbidities of obesity, respiratory disease (including asthma), diabetes, renal failure, infectious diseases, as these are recognized risk factors for severe dengue and maybe confounders (like hypertension, already “spelled-out”) for some of the outcomes you have described.

Thanks for the opportunity to review a second reiteration your manuscript.

Celia DC Christie

2 Sept, 2022

Reviewer #2: (No Response)

**Editorial and Data Presentation Modifications?**

Reviewer #1: PLOS Neglected Tropical Diseases submission (PNTD-D-22-00741)

Summary: This pilot study has shown that sST2 levels were elevated in patients older than 18 years with dengue especially in cases of severe dengue and increased sST2 levels were associated with cardiac indicators suggesting lower cardiac performance. They conclude that while sST2 maybe a useful prognostic biomarker of severe dengue, further research is needed to demonstrate its clinical utility.

Assessment: The authors have made the relevant requested corrections. The revised manuscript is now suitable for publication, with minor and other edits, viz:

Lines 19-25 and lines 26-33: Suggest separating the methods (line 19-25) from the results/principal findings (lines 26 to 33), under separate subheadings and paragraphs, for clarity.

Line 20: Suggest adding patient age for clarity, viz: “… patients aged >16 years …” 

Line 75: Suggest including “dengue” in this sentence to make the definitive link between “severe dengue, mortality and sST2”. Reference (17-18)

Lines 76-77; line 255: How old were these “younger” patients? What was/were their age range/s? References (19-22)

Line 147: Would spell out the “8 with other end-organ impairment” and how many organ systems (eg., range, median) were involved per patient, for clarity. These could have adversely contributed to the outcomes.

Line 246: Suggest removing “but”

Table 1, Results, Limitations: Given DENV endemicity in this region and the heterogenous serotypes that were co-circulating (44% positive for DENV: 57.5% DENV 2, 35.5% DENV 3 and 2.5% each for DENV 3 and DENV 4; with shift from DENV 2 to DENV 3) and dengue serotype heterogeneity with repeated infections (asymptomatic infections being the vast majority – not shown here) is a definitive risk factor for severe dengue, shouldn’t these results be “spelled out” in the narrative of the results and mentioned in the limitations (as a confounder), as this is a recognized factor risk factor for severe dengue? 

Table 1, Results: Suggest adding a footnote that there were no recognized cases with co-morbidities of obesity, respiratory disease (including asthma), diabetes, renal failure, infectious diseases, as these are recognized risk factors for severe dengue and maybe confounders (like hypertension, already “spelled-out”) for some of the outcomes you have described.

Thanks for the opportunity to review a second reiteration of your manuscript.

Celia DC Christie

2 Sept, 2022

Reviewer #2: Line 142. Consider to use different language to refer to hypertension as a preexisting condition throughout the document, as this could be confusing to your readers. Suggest to use "history of hypertension" instead of being "hypertensive", the latter may seem that you are referring to the actual blood pressure measured during disease, not the report of an existing condition. 

Line 145. Table 1. The distribution of severe dengue types add to more than 11, so I assume patients had more than one severe dengue manifestation. Consider reporting the n for those with a singular manifestation, and then using something like "XX patients had a combination of 2 or more severe dengue manifestations" 

Line 149. As recommended before, your tables should be stand alone i.e., if I read the title I can tell who are your population, when and where. Please check the titles and complete them. 

Line 158. History taking and medical records can be very unreliable for determining dengue previous infection. As you did not find any difference by this variable, consider using only serology if timing was appropriate to determine who had previous dengue infection.

Line 301. This is a very complex sentence. Consider to delete "dengue associated" or to rephrase

**Summary and General Comments**

Reviewer #1: PLOS Neglected Tropical Diseases submission (PNTD-D-22-00741)

Summary: This pilot study has shown that sST2 levels were elevated in patients older than 18 years with dengue especially in cases of severe dengue and increased sST2 levels were associated with cardiac indicators suggesting lower cardiac performance. They conclude that while sST2 maybe a useful prognostic biomarker of severe dengue, further research is needed to demonstrate its clinical utility.

Assessment: The authors have made the relevant requested corrections. The revised manuscript is now suitable for publication, with minor and other edits, viz:

Lines 19-25 and lines 26-33: Suggest separating the methods (line 19-25) from the results/principal findings (lines 26 to 33), under separate subheadings and paragraphs, for clarity.

Line 20: Suggest adding patient age for clarity, viz: “… patients aged >16 years …” 

Line 75: Suggest including “dengue” in this sentence to make the definitive link between “severe dengue, mortality and sST2”. Reference (17-18)

Lines 76-77; line 255: How old were these “younger” patients? What was/were their age range/s? References (19-22)

Line 147: Would spell out the “8 with other end-organ impairment” and how many organ systems (eg., range, median) were involved per patient, for clarity. These could have adversely contributed to the outcomes.

Line 246: Suggest removing “but”

Table 1, Results, Limitations: Given DENV endemicity in this region and the heterogenous serotypes that were co-circulating (44% positive for DENV: 57.5% DENV 2, 35.5% DENV 3 and 2.5% each for DENV 3 and DENV 4; with shift from DENV 2 to DENV 3) and dengue serotype heterogeneity with repeated infections (asymptomatic infections being the vast majority – not shown here) is a definitive risk factor for severe dengue, shouldn’t these results be “spelled out” in the narrative of the results and mentioned in the limitations (as a confounder), as this is a recognized factor risk factor for severe dengue? 

Table 1, Results: Suggest adding a footnote that there were no recognized cases with co-morbidities of obesity, respiratory disease (including asthma), diabetes, renal failure, infectious diseases, as these are recognized risk factors for severe dengue and maybe confounders (like hypertension, already “spelled-out”) for some of the outcomes you have described.

Thanks for the opportunity to review a second reiteration of your manuscript.

Celia DC Christie

2 Sept, 2022

Reviewer #2: Thanks to the authors for the modifications and answers to the questions, the manuscript reads much better with the changes.

PLOS authors have the option to publish the peer review history of their article (what does this mean?). If published, this will include your full peer review and any attached files.

Reviewer #1: No

Reviewer #2: No

Figure Files:

Data Requirements:

Reproducibility:

References

---

## [Editor Report · Decision Letter 2]

3 Oct 2022

Dear Dr Teo,

We are pleased to inform you that your manuscript 'Clinical and prognostic relevance of sST2 in adults with dengue-associated cardiac impairment and severe dengue' has been provisionally accepted for publication in PLOS Neglected Tropical Diseases.

Best regards,

Michael Cappello, MD

Academic Editor

David Harley

Section Editor

---

## [Editor Report · Acceptance letter]

8 Oct 2022

Dear Dr Teo,

We are delighted to inform you that your manuscript, "Clinical and prognostic relevance of sST2 in adults with dengue-associated cardiac impairment and severe dengue," has been formally accepted for publication in PLOS Neglected Tropical Diseases.

Best regards,

Shaden Kamhawi

co-Editor-in-Chief

Paul Brindley

co-Editor-in-Chief
